# Time- and Spectrally-Resolved Photoluminescence Study of Alloyed Cd_x_Zn_1−x_Se_y_S_1−y_/ZnS Quantum Dots and Their Nanocomposites with SPIONs in Living Cells

**DOI:** 10.3390/ijms23074061

**Published:** 2022-04-06

**Authors:** Anna Matiushkina, Ilia Litvinov, Anastasia Bazhenova, Tatiana Belyaeva, Aliaksei Dubavik, Andrei Veniaminov, Vladimir Maslov, Elena Kornilova, Anna Orlova

**Affiliations:** 1Faculty of Photonics, School of Physics and Engineering, ITMO University, 197101 Saint Petersburg, Russia; anyuta.matyushkina@mail.ru (A.M.); bazhenova.a.s@gmail.com (A.B.); adubavik@itmo.ru (A.D.); avveniaminov@itmo.ru (A.V.); vgmaslov@itmo.ru (V.M.); a.o.orlova@gmail.com (A.O.); 2Institute of Cytology, Russian Academy of Science, 194064 Saint Petersburg, Russia; tatbelyaeva@gmail.com (T.B.); lenkor@incras.ru (E.K.); 3Higher School of Biomedical Systems and Technologies, Peter the Great St. Petersburg Polytechnic University, 195251 Saint Petersburg, Russia

**Keywords:** quantum dots, SPIONs, magnetic-luminescent composites, photoluminescence, theranostics

## Abstract

Magnetic-luminescent composites based on semiconductor quantum dots (QDs) and superparamagnetic iron oxide nanoparticles (SPIONs) can serve as a platform combining visualization and therapy. Here, we report the construction of QD-SPION nanocomposites based on synthesized SPIONs and alloyed QDs $({\mathrm{Cd}}_{x}{\mathrm{Zn}}_{1-x}{\mathrm{Se}}_{y}S_{1-y})/\mathrm{ZnS}$ solubilized with L-cysteine molecules. The study of the spectral-luminescence characteristics, the kinetics of luminescence decay show the composite’s stability in a solution. After incubation with HeLa cells, QDs, SPIONs, and their composites form clusters on the cell surface and associate with endosomes inside the cells. Component-wise analysis of the photoluminescence decay of cell-associated QDs/SPIONs provides information about their localization and aggregate status.

## 1. Introduction

The development of nanotechnology makes it possible to use various materials to create nano-sized structures. Nanostructures are increasingly used in various fields, including biology and medicine, thus forming a new area of research: nanomedicine [1]. It involves the use of nanoparticles (1–100 nm) to track the behavior of labeled cells, particularly tumor cells, and to carry out targeted therapy, or a combination of both methods (“theranostics”) [2]. For these purposes, depending on the specific task, nanoparticles can be technologically designed and modified so that they simultaneously contain components for visualization and impact on transformed cells, leading to their death, for example, as a result of hyperthermia.

Over the last few decades, semiconductor nanocrystals, referred to as quantum dots (QDs), and different composites based on them have been extensively examined as luminescent labels and theranostic agents in living cells and laboratory animals [3,4,5,6]. In particular, the combination of QDs with magnetic nanoparticles is of interest.

Indeed, iron oxide nanoparticles (${\mathrm{Fe}}_{3}O_{4}$ or γ-Fe${}_{2}$O${}_{3}$) are one of the promising materials for multifunctional nanostructures [7]. Work has been conducted on the formation of multicomponent nanostructures that include, in addition to iron oxide nanoparticles, various polymeric and inorganic compounds, which allows the nanostructures to be used in various fields of biomedical research. The integrity of the resulting nanostructures is provided by binding the components using various intermolecular interactions or direct immobilization, for example, through thiol bonds, etc. [8,9,10]. Unlike macro-sized iron compounds, iron oxide nanoparticles do not have residual magnetization in the absence of an external magnetic field, which makes them superparamagnets. In connection with their properties, they are called superparamagnetic iron oxide nanoparticles (SPIONs) [11]. They can be used for magnetic hyperthermia therapy under the condition of specific interaction with tumor cells [12,13,14,15,16]. For example, antibodies specific to cancer cell receptors attached to SPIONs led to their targeting properties [17,18]. Other molecules that are characteristic of certain transformed cells, for example, the overexpressed proteins mesothelin or epidermal growth factor receptor, can also be targets [19,20,21]. SPIONs can be detected using magnetic resonance imaging (MRI), but this method is difficult to use widely.

On the other hand, QDs have unique properties for detection by fluorescence microscopy. They are characterized by a large effective Stokes shift, high quantum yield of photoluminescence (PL), narrow PL spectra, broad absorption spectra, high molar extinction coefficients, and high photostability [22]; in addition, it is possible to attach various target molecules to the QD surface. For biomedical applications, core/shell QDs are most often used (for example, QDs with a CdSe core coated with a ZnS shell). Such QDs are coated by mercaptopropionic acid, alkylated amphiphilic polymers, glutathione, PEG, L-cysteine, and other water-soluble compounds [23,24,25,26] for use in living systems.

Despite an enormous number of publications discussing QDs with different chemical compositions in regard to their biological and medical applications [27,28], the mechanisms of cellular uptake of nanoparticles, the interaction with intracellular structures, and the relaxation of electron excitation in QD-based hybrid structures at the unstable ionic composition of an intraendosomal medium in living cells remain unexplored. This is, first of all, because in most cases, only steady-state luminescence characteristics of QDs are taken into account. Similar principles are presented in the study by Zhou et al. [29]. Any information on average PL decay time can rarely be found for QDs, while one of the fundamental distinctions of QDs from organic luminophores, namely multiexponentiality of PL decay, is almost never taken into consideration [30]. Possible reasons for the multi-exponential nature of the QD PL decay are the presence of a ”dark” state [31], from which the transition to the ground state is spin-forbidden, as well as “blinking,” that is, the ability of nanocrystals to randomly switch between states with different rates of non-radiative relaxation of electronic excitation [32]. We believe that only the component-wise analysis of PL decay makes it possible to obtain realistic information on how the biological environment impacts on PL quantum yield of QDs. It allows for obtaining typical protocols for imaging analysis using QDs and for examining the stability of QD-based hybrid structures in living systems.

Here, we present a comprehensive analysis of PL properties of the commonly used CdSe-based QDs and QD-SPION magneto-luminescent composites, which are the most popular systems for tracking analysis and hyperthermia in living cells [33,34,35,36,37]. These nanostructured objects have been selected due to (i) their typical behavior in living cells, i.e., their uptake, toxicity, and intracellular localization in cells; (ii) the well-developed and specified synthetic protocols for QDs and SPIONs; and (iii) a typical approach for the composite development of QD-based hybrid structures, i.e., using stabilizing molecules, particularly L-cysteine, on the QD surface as a linker for creation chemical bonds between QDs and SPIONs.

We analyze the optical properties of QDs and QD-SPION composites in dimethyl sulfoxide (DMSO) in detail. Then, we assess uptake and intracellular localization of nanostructures in HeLa and A549 cells. Moreover, we analyze the PL kinetics of QDs and QD-SPIONs both in solvents and cells using spectrally selected single-photon counting. We show that our approach based on PL analysis of this complex system allows for correctly evaluating the average PL quantum yield of QDs as well as for clearly confirming QD aggregation and the integrity of QD-based composites in living cells.

## 2. Materials and Methods

### 2.1. Synthesis of Alloyed Quantum Dots $(Cd_{x}Zn_{1-x}Se_{y}S_{1-y})/ZnS$

In this work, we used $({\mathrm{Cd}}_{x}{\mathrm{Zn}}_{1-x}{\mathrm{Se}}_{y}S_{1-y})/\mathrm{ZnS}$ quantum dots with a gradient chemical composition synthesized by high-temperature organometallic synthesis according to the technique described in [38]. Briefly, 0.3 mM of CdO, 4.0 mM of $\mathrm{Zn}{\left(\mathrm{OAc}\right)}_{3}$, 5.5 mL of oleic acid, and 20 mL of octadecene (ODE) were placed in a 50 mL three-necked flask equipped with a magnetic stirrer and a thermometer. The reaction mixture was degassed for 1 h at a temperature of 100 °C, after which it was heated to 310 °C in an argon atmosphere. Then, 0.25 mM Zn and 3.5 mM Se were dissolved in 3 mL trioctylphosphine (TOP), quickly introduced into the reaction mixture at a temperature of 310 °C, kept for 1.5 min, and cooled in a water bath to room temperature under argon. The obtained nanocrystals were precipitated with a mixture of acetone and methanol and redissolved in chloroform (all solvents from Lenreactiv, Saint Petersburg, Russia).

Since organic solvents are not suitable for biological applications, these hydrophobic nanocrystals were transferred to DMSO using L-cysteine as a stabilizer for colloidal nanoparticles. To carry out solubilization with L-cysteine, the nanostructures were dissolved in chloroform and kept at a temperature of −20 °C for 15 min, after which a solution of L-cysteine in methanol was added until flocculation was formed. The mixture was stirred vigorously, and water and a concentrated KOH solution were added. Then, it was stirred again until about half of the nanostructures passed into water. The resulting aqueous solution of QDs $({\mathrm{Cd}}_{x}{\mathrm{Zn}}_{1-x}{\mathrm{Se}}_{y}S_{1-y})/\mathrm{ZnS}$ solubilized with L-cysteine molecule nanostructures was used for further work (all precursors from Merck KGaA, Darmstadt, Germany).

### 2.2. Synthesis of Magnetic Nanoparticles and Analysis of Samples Size

We used superparamagnetic ${\mathrm{Fe}}_{3}O_{4}$ nanoparticles obtained as a result of an original synthesis developed on the basis of techniques [39,40,41,42]. Briefly, 12 mL of triethylene glycol (TEG) and 0.5 mmol $\mathrm{Fe}{\left(\mathrm{acac}\right)}_{3}$ were placed in a two-necked flask with a volume of 50 mL equipped with a magnetic stirrer and a thermometer. The solution was degassed at 100 °C for 90 min, after which it was heated to 275 °C in an argon atmosphere and kept at this temperature for 2 h. The reaction mixture was cooled to room temperature under argon. The resulting magnetic nanoparticles were precipitated with tetrahydrofuran (THF) and dissolved in DMSO (all precursors from Merck KGaA, Darmstadt, Germany).

The SPION size was estimated using a transmission electron microscope (TEM), and their magnetic properties were investigated using a vibrating sample magnetometer. The average size of the obtained samples was estimated by dynamic light scattering (DLS) (Zetasizer Nano ZS (Malvern Panalytical, Malvern, UK)) and scanning electron microscopy (SEM) (MERLIN (Carl Zeiss, Jena, Germany)).

### 2.3. Spectral and Luminescent Properties of QDs and Their Nanocomposites

The methods of steady-state optical spectroscopy have been used to study the optical properties of samples. Absorption and PL spectra were measured on a UV-3600 spectrophotometer (Shimadzu, Kyoto, Japan) and a Cary Eclipse spectrofluorometer (Agilent, Mulgrave, Australia).

The PL kinetics of the samples was studied using a MicroTime 100 time-resolved confocal fluorescence microscope (PicoQuant, Berlin, Germany). The radiation wavelength of the pulsed diode laser was 405 nm, the pulse duration was 50 ps, and the pulse repetition frequency was 5 MHz. The PL decay curves of nanocomposites were approximated according to the following equation:(1)$$I=A_{0}+A_{1}\cdot exp\left(-\frac{t}{\tau_{1}}\right)+A_{2}\cdot exp\left(-\frac{t}{\tau_{2}}\right)+A_{3}\cdot exp\left(-\frac{t}{\tau_{3}}\right)$$
where $A_{i}$ and $\tau_{i}$ are the amplitudes at the initial time and the PL decay times of the *i*th component, respectively. The average PL lifetime is calculated using the equation: (2)$$\langle \tau \rangle =\frac{\sum_{i}A_{i}\tau_{i}^{2}}{\sum_{i}A_{i}\tau_{i}}$$

The average PL decay time was recorded using a Continuous filter monochromator b (Carl Zeiss, Jena, Germany) characterized by 10 nm full width at half maximum (FWHM) pass band at 555, 570, and 580 nm. The luminescence acquisition wavelength was chosen at the maximum of the PL spectrum as well as in the shorter-wavelength and longer-wavelength parts of the PL spectrum of QDs to obtain data from QD fractions of different sizes.

### 2.4. Cell Cultures and Compartment Identification

Human epidermoid cancer HeLa cells and human epithelial lung cancer A549 cells (Institute of Cytology, Center of collective using “Cultures collection of cells of vertebrates”, supported by Ministry of Science and Higher Education of the Russian Federation (Agreement №075-15-2021-683)) were grown in Dulbecco’s modified Eagle medium (DMEM, Biolot, St.Petersburg, Russia) with 10% fetal bovine serum (FBS, Biolot, St.Petersburg, Russia) and 1% penicillin/streptomycin (GIBCO, Waltham, Massachusetts, USA) at 37 °C with 5% ${\mathrm{CO}}_{2}$. The cells were plated on Petri dishes or Petri dishes with glass coverslips (Nunc). The experiments were performed at 60–70% confluence 48 h after seeding. To displace functional cholesterol from the membrane, the cells were pretreated with Methyl-β-cyclodextrin (MβCD) (5 mM, 2 h), followed by incubation with QDs (50 {nM ) for 24 h.

LysoTracker Green DND-26 (Invitrogen, Eugene, OR, USA) at a concentration of 50 nM was used for vital staining of lysosomes and late endosomes. After incubation with nanostructures, LysoTracker was added to the culture medium 20 min before confocal imaging. For vital staining of nuclei, Hoechst 33342 (Invitrogen, Eugene, OR, USA) was used at a concentration of 1.6 μM for 5 min. After the indicated periods, the cells were washed from dyes and analyzed.

For immunofluorescent staining of early endosomes with EEA1-antibody and lysosomes with Lamp1-antibody, the cells were fixed with 4% paraformaldehyde for 15 min, permeabilized with 0.5% Triton X-100 for 15 min, and blocked with 1% BSA for 1 h. Then, the cells were incubated for 1 h at room temperature with the selected primary antibodies against EEA1 at a dilution of 1:200 (Transduction Lab, Franklin Lakes, NJ, USA) and against Lamp-1 at a dilution of 1:100 (Abcam, Waltham, MA, USA) and for 1 h with secondary antibodies (Alexa 488 goat anti-mouse IgG, 1:500, Molecular Probes, Eugene, OR, USA). After immunostaining, the cells were plated on Fluorescent Mounting Medium (Dako Cytomation, Glostrup, Denmark) and processed for confocal imaging and, where indicated, differential interference contrast (DIC) microscopy.

### 2.5. Identification of SPION Localization by Microscopy

After incubation with the cells, SPIONs were detected with the help of Perls’ Prussian blue stain, which is a standard indicator for iron ions [43,44]. The samples were washed from unbound nanostructures using PBS and then fixed according to the standard protocol. Then, the samples were incubated in a solution of 4% HCl/4% potassium hexacyanoferrate (II) (1:1) for 20 min. The samples were washed and stained with Neutral red 0.05% dye for 20 min and, then, washed with PBS. Transmitted light microscopy images of cells with nanoparticles were obtained with a LSM 5 PASCAL microscope (Carl Zeiss, Jena, Germany).

### 2.6. Confocal Microscopy of the Cells

The cells were examined using a FV3000 inverted laser scanning confocal microscope (Olympus, Tokyo, Japan). Photoluminescence of QDs and QD-SPION nanocomposites was excited at 405 nm and recorded in the range of 520–620 nm. Alexa 488 fluorescence was excited at 488 nm and recorded in the 500–550 nm range. Hoechst 33342 fluorescence was excited at 405 nm and recorded in the range of 430–480 nm. LysoTracker Green DND-26 fluorescence was excited at 488 nm and recorded in the 500–550 nm range. The samples were observed with a 40/1.42× oil immersion objective, obtaining images of 1024 × 1024 pixels. Images were captured in one or two spectral channels in sequential scan mode, with only one laser operating at a time to avoid spectral overlap. Z-series optical sections were taken at 0.5 μm steps from bottom to top (14–16 sections).

In each experiment, 5–10 fields containing 20–60 cells totally were imaged for each experimental point. Images were processed and analyzed using Fiji software (National Institutes of Health, Bethesda, MD, USA). The most representative single sections from a Z-series of typical cells were selected for demonstration. For quantitative analysis, raw images were used.

### 2.7. Analysis of the Viability of Cell Population by Flow Cytometry

Fluorescence activated cell sorting (FACS) analysis was carried out by CytoFLEX cytometer (Beckman Coulter, Brea, CA, USA) at a maximum sample feed rate (1 μL/s) for 100 s. To study the amount of QDs associated with cells, the suspensions of control and incubated with QDs cells were analyzed with laser excitation at 405 nm; luminescence was recorded using a 660/20 BP filter. Under the same conditions, the study of the MβCD effect on the cells was carried out. The data obtained were analyzed using the CytExpert 2.0.0.152 software (Beckman Coulter, Brea, CA, USA).

To assess viability, the suspensions of control cells and cells incubated with QDs or QD-SPION composites were stained with 50 μg/ml of propidium iodide (PI, Sigma, St. Louis, MO, USA) and analyzed. Data collection was based on optical characteristics: PI fluorescence versus forwarding scattering on a logarithmic scale (FSLOG/FL4LOG). The results were expressed in the proportion of live cells (not stained with PI) relative to the total number of cells in each sample. The number of total and live control cells not incubated with nanoparticles was taken for 100%.

### 2.8. Statistical and Colocalization Analysis

Statistical data processing was performed using Microsoft Office Excel 2007 (Microsoft Corporation, Albuquerque, NM, USA). The graphs were built using the Origin 8.5 software: bar charts (mean ± standard error) and box plot—using Microsoft Office Excel 2007.

The quantitative colocalization analysis was performed using ImageJ JACoP Plugin to determine Manders’ colocalization coefficient M1, which is defined as the sum of the intensities of the selected red objects containing green signal divided by the sum of the intensities of all selected red objects. Thresholds were set by a visually estimated value for each channel. The results are represented as mean ± standard error of the mean.

## 3. Results and Discussion

### 3.1. Optical Properties of As-Prepared QDs and QD-SPION Composites in Dimethyl Sulfoxide

Appendix A shows the absorption and luminescence spectra of $({\mathrm{Cd}}_{x}{\mathrm{Zn}}_{1-x}{\mathrm{Se}}_{y}S_{1-y})/\mathrm{ZnS}$ QDs before (1) and after (2) their surface solubilization with L-cysteine. The analysis of the spectral-luminescent properties of QDs has shown that the QD luminescence band is symmetrical and has a maximum at a wavelength of 568 nm, with 26 nm FWHM. This indicates that the QDs are characterized by small size dispersion [45]. The decoration of the QD surface with L-cysteine molecules does not significantly change the position and form of their PL band. The average size of these QDs according to the evaluation of the spectral-luminescent properties is about 6 nm, which is confirmed with SEM images of the QDs (see Appendix A).

Estimation of the quantum yield of QD luminescence (ϕ) has shown that, for hydrophobic QDs before solubilization with L-cysteine, ϕ is 46% and that, after solubilization and transfer of QDs to the aqueous phase, ϕ decreased to 10.2%. A decrease in the luminescence intensity occurs due to the hole transfer from the QDs to the thiol molecule of the stabilizer. This process occurs because the trapping of holes is energetically favorable for thiols attached to QDs based on CdSe. If a hole is captured by a thiol molecule, recombination of the exciton (bound state of an electron and a hole) is impossible [46].

Briefly, the QD-SPION nanocomposites formed in an aprotonic bipolar solvent, DMSO, by the portion-wise addition of SPIONs to the QD solution in different molar ratios ($n=C_{\mathrm{SPIONs}}:C_{\mathrm{QDs}}$). The nanocomposites formed due to the coordination of L-cysteine molecules attached to the QD surface to the surface iron atoms of SPIONs [47].

Appendix A presents the optical properties of QD-SPION nanocomposites with different molar ratios (n). Appendix A shows the absorption spectra of the samples. It is clearly seen that all spectra include a specific pattern of QD absorption spectrum, i.e., the band at 552 nm.

The analysis of the PL spectra of samples (Appendix A) has shown that a sequential increase in the concentration of SPIONs in the samples is accompanied by effective quenching of the QD luminescence. The decrease in the luminescence intensity is associated both with the formation of composites and with the high absorption capacity of magnetic particles. Appendix A shows the PL spectra taking into account the effect of the internal filter associated with the high optical density of the SPIONs at the PL excitation wavelength (515 nm). Based on the quenching of luminescence, we can suppose the presence of composites in the sample. The experimental dependence of the relative intensity of QD PL on the relative concentration of SPIONs in the mixture was well approximated by the exponential function y = 0.94 · exp(−3.81x) + 0.06 (Figure 1). The exponential factor 3.81 shows that one SPION quenches the luminescence of about four QDs [48].

However, the composite formation has not been spoken of directly yet because a decrease in the quantum yield of QD luminescence can occur in the case of their aggregation, which is often observed when their immediate environment changes [49].

Therefore, we studied the PL kinetics of the QDs at different concentrations of SPIONs (Figure 2). The luminescence decay curves (Figure 2a) were approximated with the three-exponential function (see Materials and Methods, Equations (1) and (2)), the characteristic PL time and their amplitudes are presented in Table 1.

For free QDs, the characteristic PL decay times are 3.2, 13.9, and 43.9 ns. Based on the fact that the radiative rate for CdSe QDs is 25 ns [50], it can be concluded that the fraction with the decay time of 43.9 ns corresponds to the delayed luminescence of QDs [51]. The delayed luminescence component was excluded from further analysis because its PL intensity does not exceed 10% of the total PL intensity of QDs.

It is known that the PL decay time and the quantum yield of the photoluminescence of the phosphor are related according to the following formulae [52,53]:(3)$$\varphi_{i}=\frac{\tau_{i}}{\tau_{r}}$$
(4)$$\tau_{i}=\frac{1}{k_{r}+k_{nr}^{i}}$$
where $\varphi_{i}$ is the PL quantum yield of the phosphor, $\tau_{i}$ is the PL decay time, $\tau_{r}$ is the radiative time, and $k_{r}$ and $k_{nr}^{i}$ are the radiative ($k_{r}=\frac{1}{\tau_{r}}=4\cdot 10^{7}s^{-1}$ ) and non-radiative rates.

The “dark” fraction ($N_{dark}$) in the free QD solution is estimated by the following formula [54]:(5)$$\varphi =\left(1-N_{dark}\right)\cdot \varphi \left(\tau \right)$$

It equals 79%, which is explained by the high rate constant of non-radiative relaxation $k_{nr}^{1}=27\cdot 10^{7}s^{-1}$ and $k_{nr}^{2}=3\cdot 10^{7}s^{-1}$ that is comparable to or exceeds the radiative rate.

During the formation of QD-SPION composites, new channels of non-radiative QD energy relaxation are added. It is mainly associated with energy transfer from QDs to SPIONs, since the QD luminescence overlaps with the SPION absorption because they show absorption in the entire visible range at room temperature [55]. The total rate of these channels is characterized by the quenching rate of QDs with SPIONs $k_{Q}^{i}$ obtained using the following formula [56]:(6)$$\tau_{i}=\frac{1}{k_{r}+k_{nr}^{i}+k_{Q}^{i}}$$

The quenching efficiency of each QD fraction with SPIONs can be estimated by the following formula:(7)$$E_{Q}^{i}=k_{Q}^{i}\cdot \tau_{i}$$

The average quenching efficiency can be estimated taking into account the concentration of QDs in each fraction [54]: (8)$$\langle E_{Q}\rangle =\frac{\sum {\left(E_{Q}^{i}\right)}^{2}\cdot A_{i}}{\sum E_{Q}^{i}\cdot A_{i}}$$

The rate and efficiency of PL quenching of QDs with SPIONs (Table 1) rise with an increase in the SPION concentration, which means that more QDs become bound in composites with SPIONs.

It can be seen from Table 1 that an increase in the SPION concentration is accompanied by a symbate reduction in all characteristic PL decay times, which denotes the energy transfer from QD to SPION. Therefore, it confirms the formation of QD-SPION composites rather than the spontaneous aggregation of QDs, which is usually accompanied by a sharp increase in the percentage of the QD fraction with the shortest characteristic PL time [57].

No dependence of the PL characteristic decay time of QDs on the acquisition wavelength should be observed if QD ensemble is monodispersed, while QD agglomeration results in the the appearance of the PL decay time growing with the PL acquisition wavelength [58]. According to Figure 2b, the average PL decay time of nanocomposites does not depend on the acquisition wavelength within the error margin. It means the absence of energy transfer between QDs, i.e., proves the absence of their aggregation.

In two weeks, the PL decay time of QDs in the composites in water was re-measured (Appendix A), and no changes in PL kinetics of samples were found. It confirms the stability of QDs and composites.

### 3.2. Behavior and Properties of QDs and QD-SPIONs during Interaction with Cells

#### 3.2.1. Analysis of the Interaction of QD with Cultured Cells

The photophysical properties described above make it possible to detect QDs solubilized with L-cysteine molecule interactions with the cells cultivated in vitro. Figure 3 shows confocal images of HeLa and A549 cells after incubation with QDs (50 nM) for 24 h. The typical HeLa cells image focused at the level of the middle of the nucleus, and orthogonal sections of the cells, built as a result of scanning the sample along the Z-axis, correspond to the intracellular localization of QDs in vesicular endolysosomal-like structures (Figure 3a). Large QD clusters were detected in the investigated cells. However, diffuse staining of cytoplasm was negligible. To identify their localization more accurately, we combined luminescence and DIC images taken at the level of the nucleus (middle, Figure 3b) and those focused just above the highest part of the cells (top, Figure 3b). In the first case, with the focus on cytoplasm, clear vesicular structures were visible, but clusters have blurred outlines indicating their location out of focus. Several sharply defined clusters of different sizes were also detected on the coverslip outside the cells. When the focal plane was chosen just above the cells, blurred vesicular structures were seen only in DIC channel as dark dots, but clusters with clear outlines were observed. Such a picture was typical for both HeLa and A549 cells. This approach allows us to separate fluorescent objects inside and outside of rather flat cells and indicates that some number of QDs enters the cells via endocytosis while a significant portion of QDs non-specifically associates with the cell surface and forms large clusters there.

#### 3.2.2. Analysis of QD Association Dynamics with the Cells

HeLa cells incubated with QDs for 1, 4 and 24 h have been analyzed to study the dynamics of QDs interaction with the cells by combination of fluorescent microscopy and DIC imaging to separate signals from cell surface-associated and internalized QDs, as was described above. Figure 4a demonstrates that only a few small fluorescent structures can be detected both inside the cells and on their apical surface after 1 h incubation. However, in 4 h much more peripherally localized vesicular structures can be seen at the cytoplasm level, and large clusters were detected at the surface. At 24 h, the number of clusters increased and internalized structures moved into the juxtanuclear region typical for lysosomal localization. Quantitative estimation of apparent integral PL intensity per cell (Figure 4b) shows that the most intensive growth of both associated with the surface and intracellular uptake of QDs occurs between 2 and 4 h of incubation. At 4 and 24 h, a portion of internalized QDs is about twice smaller than that associated with the surface.

Such evaluations using image processing of several cells can give significant errors due to a small number (up to several dozens per experimental point) of calculated objects. To analyze a larger population (100 thousand cells), the apparent intracellular accumulation of QDs was also analyzed using flow cytometry. FACS analysis (Figure 5a) has shown that the accumulation of QDs by A549 cell line is quite similar to that by HeLa cell. The intensity distribution among the cells makes also it obvious that the accumulation of QDs by individual cells is very different. The fluorescence intensity associated with the main portion of the cells increases several times compared with the control; however, the population contains both the cells that are practically not associated with QDs and the cells in which fluorescence intensity increases by more than an order of magnitude. Additionally, some surface clusters can contribute to the total fluorescence. At the same time, the analysis of the averaged values of PL intensity per cell after 24 h of incubation shows that they are proportional to the concentration of QDs in the medium (Figure 5b), which is characteristic of fluid-phase endocytosis [59].

All types of endocytosis occur in surface membrane regions known as rafts, the main property of which is enrichment with cholesterol. Taking this into account, to quantify the ratio of internalized versus surface-associated QDs, we tried to use the incubation of cells with methyl-β-cyclodextrin (MβCD), which removes native lipid from membranes and thereby suppresses all types of endocytosis. Interestingly, this treatment led to a decrease in the luminescence intensity of cell cultures interacting with QDs (Figure 5c) almost to the control level. This result indicates that not only is the entry of QDs into the cells inhibited but also the association with the surface membrane is blocked when rafts are disturbed. The tendency to anchoring in rafts was reported earlier in the work by Karabanovas et al. [60] on QDs functionalized with carboxylic acid coating and functionalized with amino groups [61].

To track QDs’ fates upon internalization, we used classical markers of the main compartments of the endolysosomal pathway [21]. Early endosomes were identified using Alexa 488-labeled antibodies to EEA1, and lysosomes were identified using Alexa 488-labeled antibodies to Lamp1. The acidified compartments in general, which included late endosomes and lysosomes, were identified using vital staining with LysoTracker Green DND-26. Figure 6 shows the overlay of QDs luminescence channels and that of EEA1 (Figure 6a), Lamp1 (Figure 6b), and LysoTracker (Figure 6c).

Based on the data on the dynamics of QD interactions with HeLa cells (see Figure 4b), the analysis of the cargo localization in early endosomes was carried out 4 h after initiation of cells incubation with QDs. Despite the constant presence of QDs in the incubation medium, no colocalization with the marker of early endosomes EEA1 was detected (Figure 6a). Since EEA1 is a key protein in the first phase of heterotypic fusion of early endosomes [62], an event necessary for entering the pathway of lysosomal degradation, this result confirms the entry of QDs into cells via nonspecific fluid-phase endocytosis. For comparison, upon stimulation of receptor-mediated endocytosis of QDs associated with epidermal growth factor receptor (EGFR), the maximum colocalization of such targeted complexes with EEA1 reached about 70% (M1 = 0.7 ± 0.2) [21] while, in QDs, was only about 4% (M1 = 0.04 ± 0.01).

Fluid-phase endocytosis is a typically slow process without saturation effects [59]. It is well established that molecules internalized through this way are generally recycled back into the medium, but some fraction of external medium with all its compounds internalized during the formation of vesicles entering by more specific portals can reach the lysosomes. Obviously, this portion will be small enough. Indeed, in 24 h of incubation, some of the QD-containing vesicles were localized together with lysosomal markers in the juxtanuclear region of the cells (M1 = 0.27 ± 0.05 for Lamp-1 and M1 = 0.31 ± 0.07 for LysoTracker Green compared with M1 of about 0.7–0.8 in the case of targeted delivery according to our data [21]). This indicates that untargeted QDs are weakly colocalized with an acidified compartment of the degradation pathway.

Thus, the interaction of cells with the first component of the QD-SPION nanocomposites was analyzed. The study of the SPION interaction with cells is discussed in the Appendix A.

#### 3.2.3. Interaction of QD-SPION Nanocomposites with Cells

QD-SPION nanocomposites with a 1:1 molar concentration ratio of the components were incubated with HeLa cells for 24 h. Similar to experiments with free SPIONs, the presence of QD-SPIONs in HeLa cells after fixation was detected using the Pearls reaction (Figure 7a). QD-SPIONs at low concentrations (Figure 7a, left image, 25 nM) are poorly detected due to the relatively low sensitivity of the method, which is confirmed by significantly brighter and more distinct staining of cells incubated with QD-SPIONs at a concentration of 140 nM (Figure 7a, right image). However, the cytotoxicity at this concentration seems to be relatively high, since in this case, only a few cells of those initially seeded with the same density, as shown in Figure 7a (left image), remained in the field of view. These data may indicate a greater cytotoxic effect of the composites in comparison with QDs.

Nevertheless, the composites were internalized with the same efficiency as free QDs, forming on average, similar numbers of endosomes per cell. The fluorescence of QD-SPIONs was reliably detected at 20 nM (Figure 7b–d) and was localized in endosome-like structures of different sizes.

To establish more accurate localization of QD-SPIONs, cell samples with composites were analyzed using a combination of confocal microscopy and DIC (Figure 7c), as described previously. Composites, as well as in the case of single QDs and SPIONs, were found both inside the cells and on their surface. Intracellular composites were associated with relatively small vesicular structures, while there was no diffuse staining of the cytoplasm or nucleus even after 24 h of incubation. On the surface of cell membranes, composites were revealed as large clusters (Figure 7c). This arrangement of the composites is confirmed in Figure 7b, where the orthogonal projection shows a confocal image of cells with QD-SPION composites and nuclei stained with Hoechst 33342. It should be noted that in the case of composites, the colocalization of signals from nanostructures with markers of acidified endolysosomes of the degradation pathway after 24 h is rather low (M1 = 0.11 ± 0.03) (Figure 7d).

Thus, the interaction of the QD-SPION nanocomposites with cells is similar to the case of free QDs and SPIONs.

#### 3.2.4. Cytotoxicity of QDs and QD-SPION Nanocomposites

Summarizing the data obtained, it can be argued that the composites manifest themselves as stable nanostructures that are easily detected in cells and can be used in further biological research. In this regard, the study of the cytotoxic effects of both free QDs and QDs as part of the QD-SPION nanocomposites is very important. We took into account that the cytotoxicity can manifest itself in membrane damage and cell death directly on the substrate as well as in their detachment from the coverslip. In the second case, these cells are removed from the sample together with the incubation medium during sample preparation. Using flow cytometry in a series of experiments, the proportion of cells in the sample after incubation of cells with QDs or QD-SPIONs was determined in relation to control cells that were not incubated with nanoparticles (Figure 8, left columns), and among them, the proportion of live cells using the PI test was calculated (Figure 8, right column of each point).

The data obtained show no cytotoxic effect of QDs within the concentration range of 50–200 nM, since neither the total number of cells nor the proportion of viable cells noticeably differ from the control, even though the QD cores contain cadmium. We suggest that alloyed QDs reliably prevent Cd core ions from leakage into the medium. The study of the cytotoxicity of QD-SPIONs in the concentration range of 10–140 nM showed that, starting from 25 nM, a significant cytotoxic effect is observed. Thus, the cytotoxicity of QD-SPION composites is higher than the cytotoxicity of free QDs. These data confirm the changes noted in cell morphology in the presence of both free SPIONs and their composites with QDs.

### 3.3. Analysis of PL Kinetics of QDs and QD-SPIONs Using Spectrally Selected Single-Photon Counting

The widespread approach to the study of the PL properties of QDs associated with cells by steady-state luminescence methods can give an erroneous result, since the QD PL intensity can decrease for many reasons, including QD aggregation and the emergence of new non-radiative relaxation channels. We also showed earlier that the pH of the medium also affects the PL intensity of QDs [63,64]. Steady-state methods cannot answer what exactly the reason for the quenching of QD luminescence in cells is. However, additional information on the interaction of QDs with cells can be obtained by studying the QD PL decay kinetics.

The cells cultured in vitro are grown in media that mimic the biological environment of cells. Ions, vitamins, essential amino acids, proteins, and other compounds contained in such complex solutions (Thermo Fisher Scientific; DMEM, high glucose, “Technical Resources—Media Formulations” https://thermofisher.com/ru/ru/home/technical-resources/media-formulation.8.html, accessed on 15 December 2021) can affect the spectral-luminescent properties of QDs due to their interaction with the QD surface. In our experiments, HeLa cells grown on the glass coverslips were incubated in DMEM containing QDs and QD-SPION composites. After cell fixation, areas of 512 × 512 pixels, each containing about 4–5 cells, were analyzed using the PicoQuant microscope. In this experiment, we technically were not able to distinguish between luminescence from inside the cells and that from the cell surface, so the resulting luminescence lifetimes represent averaged values of signals collected from the whole cells. The QD PL lifetimes were also calculated using the model of the QD PL decay (Equations (1) and (2)). Figure 9 shows the dependence of the average PL decay time of free QDs and those bound with SPION QDs on the acquisition wavelength for different times of incubation with HeLa cells.

The average PL decay time of QDs in cells is noticeably shorter than in the solution. This is typical for all samples and is explained by the emergence of new non-radiative energy relaxation channels due to the interaction of the QD surface with the cellular environment.

Figure 9 demonstrates that, for all experiments, an increase in the average QD PL decay time with the acquisition wavelength is observed. This implies the increasing energy transfer from smaller QDs to larger QDs and, therefore, indicates the QD aggregation during their incubation with cells. The efficiency of the energy transfer process between QDs can be estimated by the following formulae:(9)$$E_{FRET}^{QD-QD}=1-\frac{\tau_{555}}{\tau_{580}}$$
(10)$$E_{FRET}^{QD-QD}=\frac{R_{0}^{6}}{R_{0}^{6}+R^{6}}$$
where $\tau_{555}$ and $\tau_{580}$ are the average PL decay times acquired at 555 and 580 nm, *R* is the average distance between quantum dots, and $R_{0}$ is the Förster radius calculated using the following formula:(11)$$R_{0}^{6}=\frac{9000\cdot ln10\cdot \Phi^{2}\cdot \varphi }{128\cdot \pi^{5}\cdot n^{4}\cdot N_{a}}\cdot J$$
where Φ is the orientation factor that takes into account the mutual orientation of the transition dipole moments in the donor and acceptor (in solution, $\Phi^{2}$ = 2/3), φ is the PL quantum yield of the energy donor, n is the refractive index of the solvent, $N_{a}$ is the Avogadro number, and *J* is the overlap integral between a donor luminescence and an acceptor absorption spectra.

An estimation of the average distance between quantum dots in cells makes it possible to calculate the energy transfer rate between QDs using the following formula:(12)$$k_{FRET}^{QD-QD}=\frac{R_{0}^{6}}{\tau_{580}\cdot R^{6}}$$

It is clear from Figure 9 and Table 2 that FRET efficiency between adjacent QDs ($E_{FRET}^{QD-QD}$) increases with incubation time due to the QD aggregation on the cell membrane. This tendency has already been detected in the analysis of confocal images (Figure 4b), where “top” corresponds to the surface (membrane) of cells and “middle” refers to inside of cells. However, upon reaching a certain value of about 30%, QD aggregation ends. This is typical for both free and composite-bound QDs.

The increased energy transfer rate in cells between composite-bound QDs, compared with free QDs, may indicate more pronounced composites’ aggregation than that of free QDs in cells.

Additionally, all incubation times are characterized by a decrease in the average PL decay time upon passing from free QDs to composites. This directly indicates the transfer of energy from QDs to SPIONs, that is, their interaction in the composites. The efficiency of the QD-SPION quenching can be estimated according to the following formula:(13)$$E_{Q}^{QD-SPION}=1-\frac{\tau_{QD-SPION}}{\tau_{QD}}$$
where $\tau_{QD}$ and $\tau_{QD-SPION}$ are the average PL decay times of free and composite-bound QDs.

Figure 10 shows that the efficiency of QD luminescence quenching in QD-SPION composites plotted as a function of incubation time saturates before 24 h of incubation regardless of the wavelength and approaches the PL quenching efficiency for these composites in solution (dashed line in Figure 10; the estimates are based on the data given in Figure 2b). This confirms the stability of the composites in cells.

A completely different picture is observed at a shorter time (4 h) of incubation of cells with QD-SPION composites: (i) the efficiency of QD PL quenching depends on the PL detection wavelength, essentially on QD size; (ii) the efficiency of QD PL quenching by SPIONs in cells is significantly lower than for 24 and 48 h in cells or for a colloidal solution of composites. It is known that the formation of aggregates of QDs or aggregates of composites comprising QDs is accompanied by the appearance of a dependence of the characteristic QD luminescence decay times on the recording wavelength. This phenomenon is attributed to the size dispersion of QDs in an ensemble, which in turn leads to the appearance of Förster energy transfer from smaller to larger QDs. The efficiency of this process can be estimated using formula (9).

As a rule, aggregation of QDs or their composites leads not only to the wavelength dependence of the PL decay but also to a sharp decrease in the luminescence quantum yield due to the appearance of effective non-radiative relaxation of QD electronic excitation. Therefore, the wavelength dependence of QD luminescence quenching and the weaker effect of SPIONs on the PL of QDs in the case of short incubation implies that most of the composites form aggregates on the cell surface. A longer incubation enables QD-SPION composites to penetrate into the cytoplasm. This leads to an increase in the average distance between the composites and, as a result, to the elimination of Förster energy transfer between QDs belonging to different composites, hence the disappearance of the non-radiative photoexcitation energy relaxation channel competing with the energy transfer from QDs to SPIONs in a composite.

## 4. Conclusions

QDs are known for their photostability and brightness, while magnetic nanoparticles can be used for hyperthermia, one of the ways to kill cancer cells. In this work, QD-SPION nanocomposites were created for the combined possibility of detection and effect on cells. The QD-SPION composites created showed fairly high stability of their properties. The pathways of cellular uptake of nanostructures were studied in detail.

It was shown that QDs and non-targeted QD-SPION nanocomposites can enter the cells by endocytosis and, during prolonged incubation, accumulate in vesicular structures, most likely in endolysosomes. It is important to note that, due to the spectral-luminescent properties of QDs, the composites are perfectly detected in cells even at low nanomolar concentrations, which allows for minimizing the risks of cytotoxicity.

The spectrally resolved analysis of the QD PL kinetics both free and as part of composites with energy transfer allows for (1) understanding the reasons for the quenching of QD luminescence in cells; (2) evaluating the average distance between QDs nearby and, on the basis of this, concluding the degree of QD aggregation in cells; and (3) drawing conclusions about the preservation/non-preservation of the QD-quencher composites integrity in cells. Therefore, the analysis of the luminescence kinetics showed that QDs and SPIONs remain bound in composites during incubation with cells. Thus, the approach based on PL decay analysis is more informative for estimations of the biological microenvironment impact on the luminescent properties of quantum dots in nanocomposites compared with the analysis of fluorescence intensity and allows us to avoid incorrect interpretation of image analysis data.

## Figures and Tables

**Figure 1 ijms-23-04061-f001:**
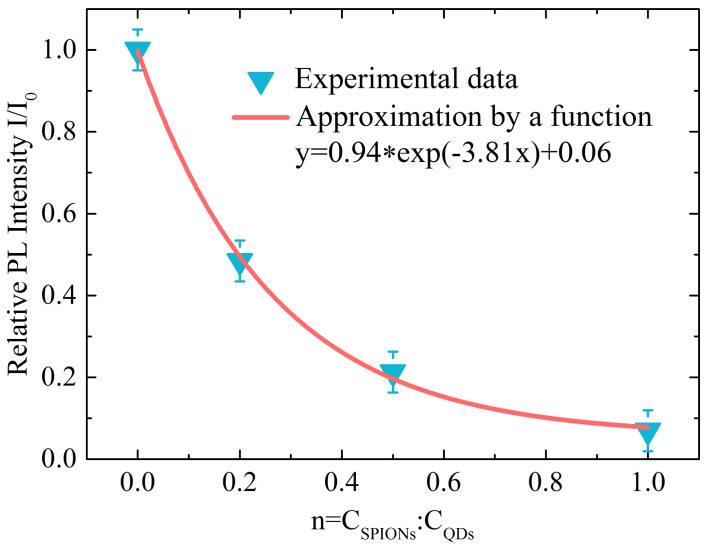
Dependence of the relative PL intensity of the samples in DMSO solution on the ratio of the molar concentrations of $C_{\mathrm{SPIONs}}/C_{\mathrm{QDs}}$. The experimental data are approximated by the function y = 0.94 · exp(−3.81x) + 0.06.

**Figure 2 ijms-23-04061-f002:**
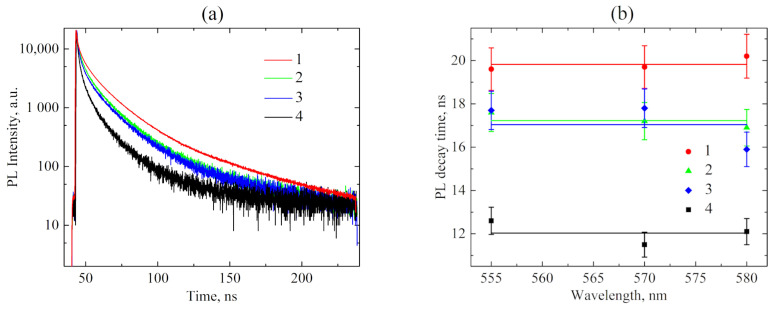
PL kinetics of QD-SPION nanocomposites. (**a**)—PL decay curves of the samples; (**b**)—Dependence of the sample average PL decay times calculated using formula (2) on the acquisition wavelength; 1—free QDs (red), 2–4—mixture (SPIONs/QDs) with n ($C_{\mathrm{SPIONs}}:C_{\mathrm{QDs}}$) = 0.2 (green); 0.5 (blue); 1 (black), samples in DMSO solution.

**Figure 3 ijms-23-04061-f003:**
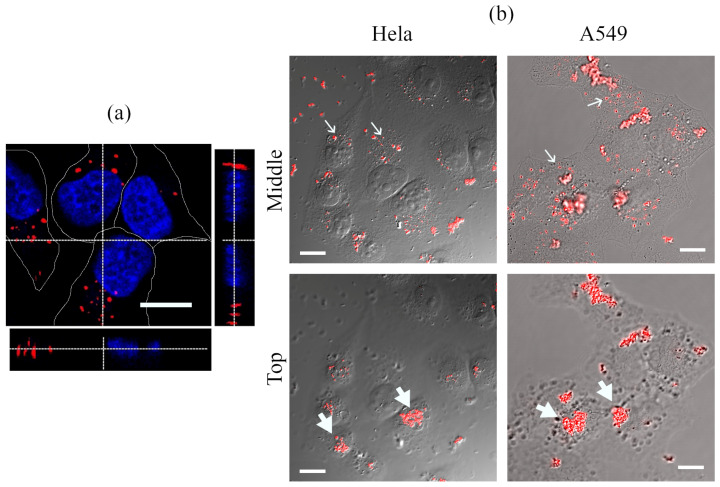
Confocal images of cells after incubation with QDs (50 nM) for 24 h. (**a**) Optical slice of HeLa cells at the nucleus level (red channel—QDs, blue channel—Hoechst 33342); below and on the right, there are orthogonal sections of cells obtained as a result of scanning along the XZ and YZ axes. (**b**) Overlay of DIC images of cells and the QD luminescence channel. Localization of QDs is represented by optical sections taken from Z-series at the level of the nucleus (**middle**) and just above the apical surface (**top**) of HeLa and A549 cells. Thin arrows show examples of QDs inside cells; thick arrows indicate of QDs association with the cell surface outside the cell. Photoluminescence of QDs excited at 405 nm and recorded in the range of 520–620 nm. Hoechst 33342 fluorescence was excited at 405 nm and recorded in the range of 430–480 nm. Scale bars 15 μm.

**Figure 4 ijms-23-04061-f004:**
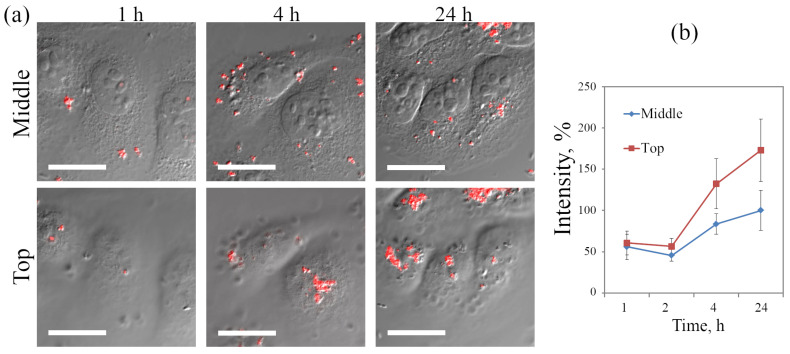
Dynamics of accumulation of QDs by HeLa and A549 cells, obtained by analysis of confocal images. (**a**) Confocal image of HeLa cells after culturing for 1, 4, and 24 h with QDs (50 nM). Typical optical sections from the Z-series at the level of the nucleus (**middle**) and on the apical surface (**top**) of cells; overlay of differential contrast images of cells and the QD luminescence channel are shown. (**b**) QDs’ apparent accumulation by HeLa cells during incubation for 1, 2, 4, and 24 h, plotted from the integral intensity of QD luminescence at the level of the nucleus (**middle**) and on the apical surface (**top**) of cells. The integral luminescence intensity of QDs per cell in the middle focal plane after 24 h of incubation is taken at 100%. Photoluminescence of QDs excited at 405 nm and recorded in the range of 520–620 nm. Scale bars 15 μm.

**Figure 5 ijms-23-04061-f005:**
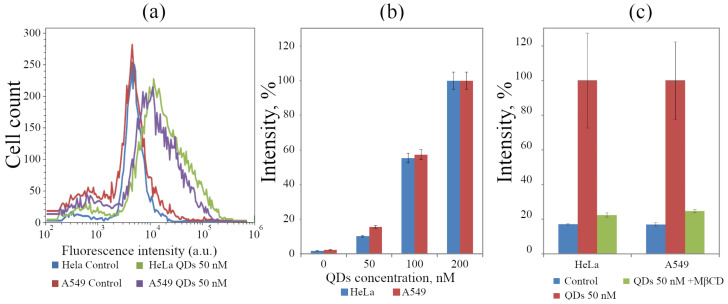
Dynamics of the accumulation of QDs by HeLa and A549 cells obtained by flow cytometry analysis. (**a**) Comparison of the distribution of the luminescence intensity of HeLa and A549 cells in the control and after incubation with 50 nM QDs for 24 h. (**b**) Concentration dependence of QD association with HeLa and A549 cells in the control and after incubation with QDs at different concentrations (50–200 nM) for 24 h. (**c**) FACS analysis of QD accumulation (50 nM, 24 h) by HeLa and A549 cells saturated with Methyl-β-cyclodextrin. The results are presented as a mean ± standard error of at least 40 cells for each case. Three independent experiments were carried out.

**Figure 6 ijms-23-04061-f006:**
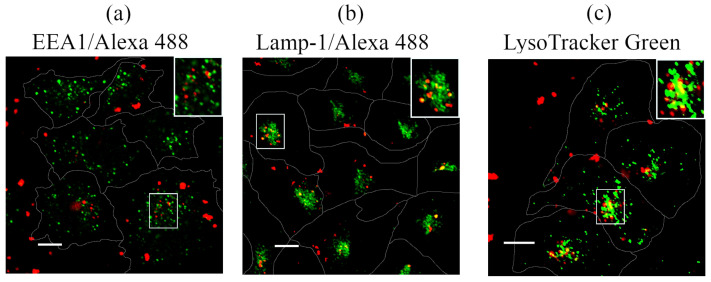
Analysis of the association of QDs with markers of the endocytic pathway. Confocal images of HeLa cells after incubation with QDs (50 nm) for 4 h (**a**) and 24 h (**b**,**c**). Cells were fixed and immunostained with EEA1/Alexa 488 (**a**) or Lamp-1/Alexa 488 (**b**) antibodies before confocal microscopy. Live cells stained with LysoTracker Green DND-26 (**c**) for 20 min before confocal imaging. The insets represent enlarged views (3×) of the corresponding boxed region of the cell. Photoluminescence of QDs excited at 405 nm and recorded in the range of 520–620 nm. Alexa 488/LysoTracker Green fluorescence was excited at 488 nm and recorded in the 500–550 nm range. Scale bars 10 μm.

**Figure 7 ijms-23-04061-f007:**
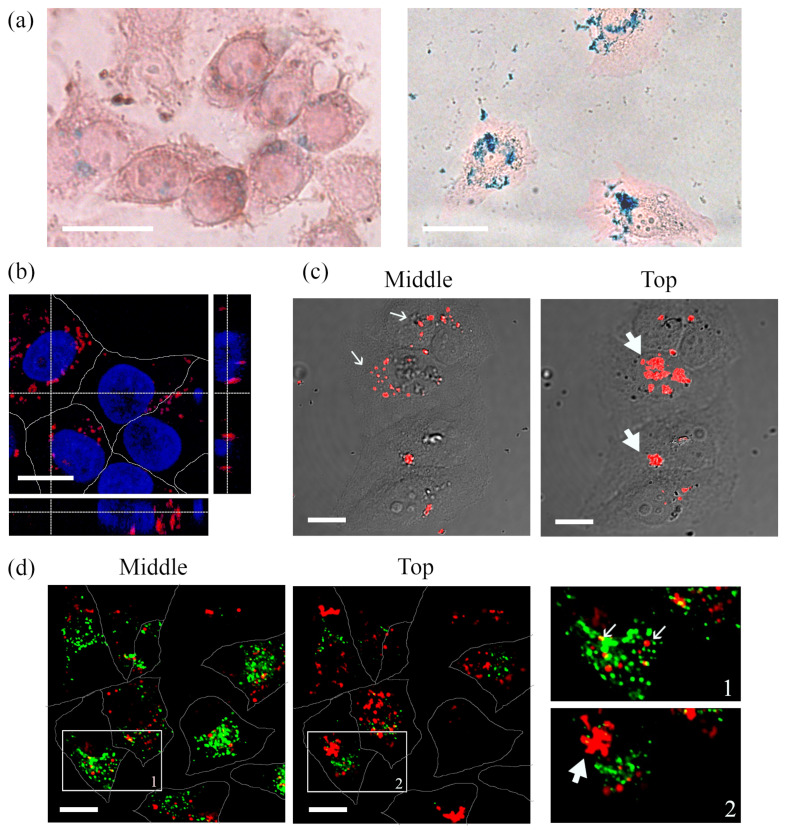
Microscopy of HeLa cells after incubation with QD-SPIONs (1:1 concentration ratio of QDs and SPIONs) for 24 h. (**a**) Light microscopy of cells stained with Perls Prussian blue to detect iron compounds after the incubation of cells with QD-SPION nanocomposites at concentrations of 25 nM (left) and 140 nM (**right**). (**b**–**d**) Confocal images of cells after incubation with QD-SPIONs (20 nM): (**b**) Optical slice of HeLa cells at the nucleus level (red channel—QD-SPIONs, blue channel—Hoechst 33342); below and on the right, there are orthogonal sections of cells obtained as a result of scanning along the XZ and YZ axes. (**c**) Overlay of DIC images of cells and the QD-SPION luminescence channel. Localization of QD-SPIONs represented by optical sections from the Z-series at the level of the nucleus (middle) and on the apical surface (**top**) of HeLa cells. (**d**) Live cells stained with LysoTracker Green DND-26 for 20 min before confocal imaging. Visualization is represented by optical sections from the Z-series at the level of the nucleus (middle) and on the apical surface (**top**) of the cells. The insets 1 and 2 represent enlarged views (3×) of the corresponding boxed region of the cell. Thin arrows show examples of QD-SPIONs inside cells; thick arrows indicate the accumulation of QD-SPIONs on the cell surface. Photoluminescence of QDs excited at 405 nm and recorded in the range of 520–620 nm. Hoechst 33342 fluorescence was excited at 405 nm and recorded in the range of 430–480 nm. LysoTracker Green fluorescence was excited at 488 nm and recorded in the 500–550 nm range. Scale bars 15 μm.

**Figure 8 ijms-23-04061-f008:**
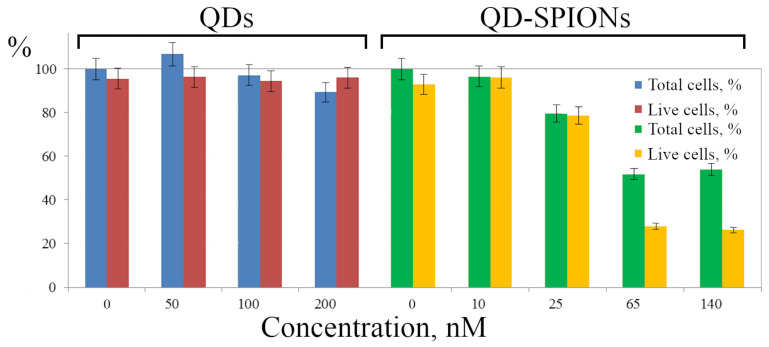
Analysis of HeLa cell viability during incubation with QDs and QD-SPIONs for 24 h. Data on the number of cells in the sample and the proportion of live cells (cell viability) are presented in %. The number of cells of control cells (no QDs or QD-SPIONs) was taken as 100%. The results represent a mean value ± standard error of the mean.

**Figure 9 ijms-23-04061-f009:**
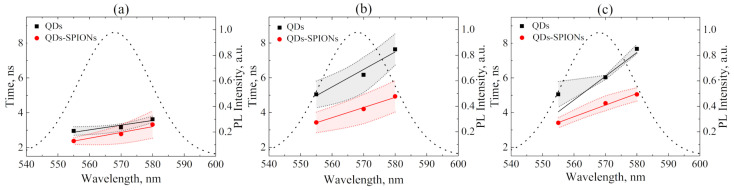
Dependence of the average PL decay time of QDs and QD-SPION nanocomposites on the acquisition wavelength for 4 (**a**), 24 (**b**), and 48 (**c**) hours of incubation in HeLa cells; the dotted curve shows the PL spectrum of QD solution in DMSO.

**Figure 10 ijms-23-04061-f010:**
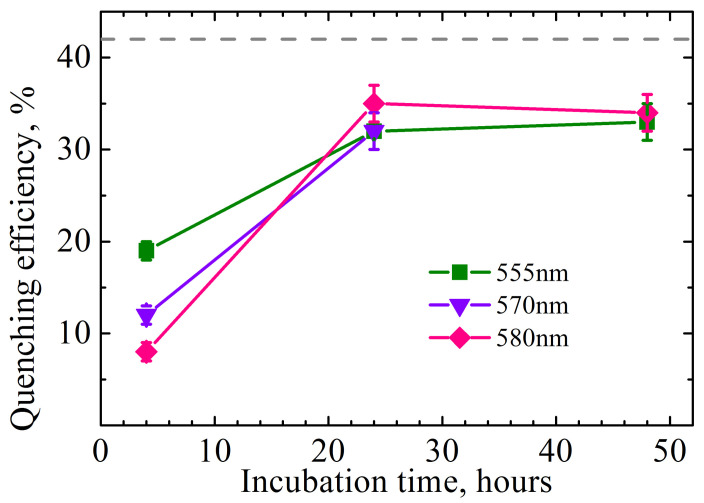
The efficiency of QD PL quenching by SPIONs in cells estimated by the PL decay times measured at 555, 570, and 580 nm, as a function of incubation time (data from Figure 9); the dashed line is PL quenching efficiency in solution (data from Figure 2b).

**Table 1 ijms-23-04061-t001:** Photophysical properties of QDs and QD-SPION nanocomposites *.

Molar Concentration Ratio ($n=C_{\mathrm{SPIONs}}:C_{\mathrm{QDs}}$)	0	0.2	0.5	1
$\tau_{1}$, ns	3.2 ± 0.2	2.4 ± 0.1	2.0 ± 0.1	1.4 ± 0.1
$A_{1}$,%	41 ± 2	49 ± 3	51 ± 3	61 ± 3
$\tau_{2}$, ns	13.9 ± 0.7	11.7 ± 0.6	10.6 ± 0.5	7.0 ± 0.4
$A_{2}$,%	51 ± 3	42 ± 2	40 ± 2	33 ± 2
$\tau_{3}$, ns	44 ± 2	34 ± 2	31 ± 2	24 ± 1
$A_{3}$,%	8 ± 1	9 ± 1	9 ± 1	6 ± 1
<τ>, ns (2)	21 ± 1	17.9 ± 0.9	16.7 ± 0.8	11.7 ± 0.6
$\varphi_{1}$,% (3)	12.8 ± 0.6	9.6 ± 0.5	8.0 ± 0.4	5.6 ± 0.3
$\varphi_{2}$, % (3)	56 ± 3	47 ± 2	42 ± 2	28 ± 1
<φ>, %	49 ± 2	40 ± 2	36 ± 2	22 ± 1
$k_{nr}^{1}$, $\cdot 10^{7}s^{-1}$ (4)	27 ± 1			
$k_{nr}^{2}$, $\cdot 10^{7}s^{-1}$ (4)	3.2 ± 0.2			
$k_{Q}^{1},\cdot 10^{7}s^{-1}$ (6)	-	10.4 ± 0.5	18.8 ± 0.9	40 ± 2
$k_{Q}^{2},\cdot 10^{7}s^{-1}$ (6)	-	1.4 ± 0.1	2.2 ± 0.1	7.1 ± 0.4
$<k_{Q}>,\cdot 10^{7}s^{-1}$	-	9.5 ± 0.5	17.4 ± 0.9	37 ± 2
$E_{Q}^{1}$, % (7)	-	25 ± 1	38 ± 2	56 ± 3
$E_{Q}^{2}$, % (7)	-	16 ± 1	24 ± 1	50 ± 3
$<E_{Q}>$, % (8)	-	22 ± 1	33 ± 2	54 ± 3

* The numbers (2)–(8) correspond to the formulas that are used to estimate these values.

**Table 2 ijms-23-04061-t002:** Parameters of the QD–QD interaction for free QDs and QD-SPION nanocomposites in cells *.

Incubation Time, h	Nanostructures Type	$\tau_{555}$, ns	$\tau_{570}$, ns	$\tau_{580}$, ns	$E_{FRET}^{\mathrm{QD}-\mathrm{QD}}$, % (9)	$R_{0}$, nm (11)	*R*, nm (10)	$k_{FRET}^{\mathrm{QD}-\mathrm{QD}},$$\cdot 10^{7}s^{-1}$ (12)
4	QDs	3.0 ± 0.2	3.2 ± 0.2	3.6 ± 0.2	18 ± 1	4.6 ± 0.2	5.9 ± 0.3	6.2 ± 0.3
4	QD-SPIONs	2.4 ± 0.2	2.8 ± 0.5	3.3 ± 0.8	28 ± 2		5.4 ± 0.3	11.9 ± 0.6
24	QDs	5.0 ± 0.8	6.2 ± 1.0	7.6 ± 0.9	34 ± 2		5.1 ± 0.3	6.7 ± 0.3
24	QD-SPIONs	3.4 ± 0.6	4.2 ± 0.8	4.9 ± 0.9	30 ± 2		5.3 ± 0.3	8.9 ± 0.4
48	QDs	5.1 ± 0.7	6.0 ± 0.1	7.7 ± 0.3	34 ± 2		5.1 ± 0.3	6.7 ± 0.3
48	QD-SPIONs	3.4 ± 0.3	4.5 ± 0.4	5.1 ± 0.4	32 ± 2		5.2 ± 0.3	9.5 ± 0.5

* The numbers (9)–(12) correspond to the formulae that are used to estimate these values.

## Data Availability

The data presented in this study are available from the authors upon request.

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
