# Peer review of "Time- and Spectrally-Resolved Photoluminescence Study of Alloyed CdxZn1−xSeyS1−y/ZnS Quantum Dots and Their Nanocomposites with SPIONs in Living Cells"

_ijms, 2022, doi:10.3390/ijms23074061_

Round 1

Reviewer 1 Report

The work entitled

Time- and spectrally-resolved photoluminescence study of alloyed CdxZn1−xSeyS1−y/ZnS quantum dots and their nanocomposites with SPIONs in living cells

by A. Matiushkina et al.

is a systematic investigation of the title system. Although the systems are well known and intensely studied, the authors added new interesting clues, somewhat incremental contributions, but yet important.  Given the invested endeavor, the article is worth of publishing, almost as it is.

However, we would suggest certain minor amendments. We think is better to move the Appendix in a Supporting Information file and, possibly, to move certain picture frames from the main text in this supplementary material.

The figures are not well arranged. For instance, Fig.1 and 3 are small graphs with a lot of white space wasted around. These can be enlarged. The figures composed from odd number of panels are not a good formatting. We suggest compositions with even number of snippets, realizing a wise covering of dedicated figure surface, once again, without empty spaces. The figure 5 is overloaded, with many panels of different nature. Reorganize it in two figures and consider to move a part in the supplementary file. Even though in electronic format is possible to zoom in, it is desirable for a figure to have the details visible at one synoptic glance.

The minor grammar issues are solvable in the final proof processing, with the help of journal staff.

Except these minor editing aspects, we consider the scientific content sound and clear.

Author Response

We are very grateful for your valuable comments and appreciate your inputs that definitely help to improve our manuscript.

Below are our point-by-point responses to the Reviewer’s comments:

Point 1: We think is better to move the Appendix in a Supporting Information file and, possibly, to move certain picture frames from the main text in this supplementary material. The figures are not well arranged. For instance, Fig.1 and 3 are small graphs with a lot of white space wasted around. These can be enlarged. The figures composed from odd number of panels are not a good formatting. We suggest compositions with even number of snippets, realizing a wise covering of dedicated figure surface, once again, without empty spaces. The figure 5 is overloaded, with many panels of different nature. Reorganize it in two figures and consider to move a part in the supplementary file. Even though in electronic format is possible to zoom in, it is desirable for a figure to have the details visible at one synoptic glance.

Response 1: Thank you very much for your significant comments on the figure presentation. We have moved some figures to the Supplementary file and enlarged the graphs â„– 1 and â„–3 according to your recommendation. Also, we have done our figures more symmetrical. We agree that Fig. 5 was overloaded, but since it is very important for the description and contains priority data, we just divided it into two parts - Fig. 4 and Fig. 5 and kept the both in the main text. We tried to make individual details of the figures more visible.

Reviewer 2 Report

Matiushkina and coworkers present an interesting analysis of the PL proprieties of fluoro-magnetic system composed by QD and SPION complexes. The investigation presented I detailed, both in lab and in vitro and the materials show a very good stability in the biological environment. Therefore, I would suggest the publication of the manuscript once the following points will be addressed:

General comments:

- Are you sure that the term “nanocomposite” is correct to describe the QD-SPION diads? With nanocomposite usually indicates polymeric host that contain nanosized material that gives them specific functionalities, which is not the case here.

- More details in the captions need to be reported, to help the reader to better understand the showed data, especially on photoluminescence.

- Line 258: During the formation of QDs-SPIONs composites, new channels of nonradiative QD 258 energy relaxation are added, which are associated with energy transfer from QDs to SPIONs

 The PL data showed indicates an effective quenching of the QD luminescence as function of the SPION concentration, thus with the increased of QD-SPION complexes. In general, this can generate quenching of the QD emission, but there no hint of energy transfer in the reported data. It can happen of course, but first of all there should be some resonance between the QD emission and SPION absorption spectra, that I cannot see. Most probably some charge transfer process occurs between the close packed species, but without any further evidence the author should not ascribe t quenching to energy transfer.

Similarly at line 450:

Fig.9 demonstrates that, for all experiments, an increase in the average QD PL decay 450 time with the acquisition wavelength is observed. This implies the increasing energy 451 transfer from smaller QDs to larger QDs, therefore indicates the QD aggregation during 452 their incubation with cells.

If this assumption is true, we should observe some redshift in the QD PL maximum, since large QD are excited by smallest QD obtaining a redshifted global emission. I cannot see it clearly in the spectra reported in Fig.9. The reported data suggest that in the biological environment some other quenching process is active, but I feel difficult to assign it to the process descried, unless more data demonstrate it.

A couple of relevant references must be added

Advanced Functional Materials 28 (19), 1707582

Nano Lett. 2011,11, 3404.

ACS Nano 2011, 5, 8230.

Author Response

We are very grateful for your valuable comments and appreciate your inputs that definitely help to improve our manuscript.

Below are our point-by-point responses to the Reviewer’s comments:

Point 1: Are you sure that the term “nanocomposite” is correct to describe the QD-SPION diads? With nanocomposite usually indicates polymeric host that contain nanosized material that gives them specific functionalities, which is not the case here.

Response 1: You are right and indeed the term nanocomposites implies the presence of organic compounds. Nevertheless, there are works in which nanocomposites are also understood as structures that include inorganic fluorophores, such as quantum dots. Therefore, in this study, we use this term to emphasize the multicomponent nature of the system. Below we provide examples of the use of this term in articles.

You, Xiaogang, et al. "Hydrophilic high-luminescent magnetic nanocomposites." Nanotechnology 18.3 (2007): 035701. (https://doi.org/10.1088/0957-4484/18/3/035701) «…simple method to fabricate hydrophilic, highly luminescent magnetic nanocomposites (LMNCs) based on the connection of QDs flocculation and MNPs through charge interaction…»

Quarta, Alessandra, et al. "Fluorescent-magnetic hybrid nanostructures: preparation, properties, and applications in biology." IEEE Transactions on Nanobioscience 6.4 (2007): 298-308. (https://doi.org/10.1109/TNB.2007.908989) «…hybrid nanocomposites having either organic dye molecules or quantum dots as fluorescent domains, and iron oxide nanoparticles as the magnetic domains…»

Ma, Zhi Ya, et al. "Synthesis and bio-functionalization of multifunctional magnetic Fe 3 O 4@ Y 2 O 3: Eu nanocomposites." Journal of materials chemistry 19.27 (2009): 4695-4700. (https://doi.org/10.1039/B901427F) «…Most of the magnetic fluorescent nanocomposites are core–shell structures with the great majority of emitters being either quantum dots (QDs) or organic dyes…»

Point 2: More details in the captions need to be reported, to help the reader to better understand the showed data, especially on photoluminescence.

Response 2: Your remark is very important, therefore, in order to improve the perception of images, we added data on the excitation of luminescence and the emission ranges of QDs and the fluorophores used in the captions to the figures and other clarifying information.

Point 3: Line 258 (263 NEW file): During the formation of QDs-SPIONs composites, new channels of nonradiative QD energy relaxation are added, which are associated with energy transfer from QDs to SPIONs

The PL data showed indicates an effective quenching of the QD luminescence as function of the SPION concentration, thus with the increased of QD-SPION complexes. In general, this can generate quenching of the QD emission, but there no hint of energy transfer in the reported data. It can happen of course, but first of all there should be some resonance between the QD emission and SPION absorption spectra, that I cannot see. Most probably some charge transfer process occurs between the close packed species, but without any further evidence the author should not ascribe t quenching to energy transfer.

Response 3: Thank you for bringing this matter to attention, though in the general case, both processes are possible in this system - energy transfer, since the QD luminescence overlaps with the SPION absorption, because they absorb in the entire visible range, and charge transfer. However, in our particular case, when QDs and SPIONs are interconnected via a stabilizer molecule, the charge transfer efficiency will be significantly lower than the energy transfer efficiency. Therefore, energy transfer is considered in this work as the dominant mechanism of QD luminescence quenching.

In this regard, following text is inserted into the article:

Line 263: «During the formation of QDs-SPIONs composites, new channels of nonradiative QD energy relaxation are added. It is mainly associated with energy transfer from QDs to SPIONs, since the QD luminescence overlaps with the SPION absorption, because they absorb in the entire visible range at room temperature [55

Point 4: Similarly at line 450 (457 NEW):

Fig.9 demonstrates that, for all experiments, an increase in the average QD PL decay time with the acquisition wavelength is observed. This implies the increasing energy transfer from smaller QDs to larger QDs, therefore indicates the QD aggregation during their incubation with cells.

If this assumption is true, we should observe some redshift in the QD PL maximum, since large QD are excited by smallest QD obtaining a redshifted global emission. I cannot see it clearly in the spectra reported in Fig.9. The reported data suggest that in the biological environment some other quenching process is active, but I feel difficult to assign it to the process descried, unless more data demonstrate it.

Response 4: We completely agree that in the general case, a red shift of the PL QD maximum should be observed. However, this shift can be insignificant, especially if both free and aggregate-included QDs contribute to the luminescence spectrum. In this work, we studied the QD luminescence kinetics in cell cultures. The specificity of the samples does not allow recording luminescence spectra in the cells with the required spectral resolution. The conclusion about the aggregation of nanostructures in cell cultures was made on the basis of the QD luminescence kinetics data, which clearly demonstrates the dependence of the average decay time on the acquisition wavelength in the range of the exciton band of the QD luminescence. The luminescence spectrum shown in these figures corresponds to the QD luminescence spectrum in solution and serves to visually explain the choice of acquisition wavelengths. We indicate this detail in the new figure (Fig.9) caption.

Point 5: A couple of relevant references must be added: Advanced Functional Materials 28 (19), 1707582; Nano Lett. 2011,11, 3404.; ACS Nano 2011, 5, 8230.

Response 5: These references are extremely interesting and therefore we have included them as refs [8-10] in the Introduction with appropriate comment.

Line 27: «There are works on the formation of multicomponent nanostructures that include, in addition to iron oxide nanoparticles, various polymeric and inorganic compounds, which allows the nanostructures to be used in various fields of biomedical research. The integrity of the resulting nanostructures is provided by binding the components using various intermolecular interactions or direct immobilization, for example, through thiol bonds, etc.[8-10]»

Reviewer 3 Report

The manuscript is clear, well arranged, the experimental and theoretical results are clearly presented.

The research is scientifically sound, original, the motivation is clear and the results may have impact in the for bioimaging applications related to the luminescent properties of quantum dots in nanocomposites

Author Response

We are very grateful for your appreciation of our work.